# PhysGNN: A Physics–Driven Graph Neural Network Based Model for Predicting Soft Tissue Deformation in Image–Guided Neurosurgery

**Yasmin Salehi,  Dennis Giannacopoulos**
Department of Electrical and Computer Engineering
McGill University
Montreal, QC, Canada
`yasmin.salehi@mail.mcgill.ca, dennis.giannacopoulos@mcgill.ca`

## Abstract

Correctly capturing intraoperative brain shift in image-guided neurosurgical proce-
dures is a critical task for aligning preoperative data with intraoperative geometry
for ensuring accurate surgical navigation. While the finite element method (FEM)
is a proven technique to effectively approximate soft tissue deformation through
biomechanical formulations, their degree of success boils down to a trade-off
between accuracy and speed. To circumvent this problem, the most recent works
in this domain have proposed leveraging data-driven models obtained by train-
ing various machine learning algorithms—e.g., random forests, artificial neural
networks (ANNs)—with the results of finite element analysis (FEA) to speed up
tissue deformation approximations by prediction. These methods, however, do
not account for the structure of the finite element (FE) mesh during training that
provides information on node connectivities as well as the distance between them,
which can aid with approximating tissue deformation based on the proximity of
force load points with the rest of the mesh nodes. Therefore, this work proposes a
novel framework, PhysGNN, a data-driven model that approximates the solution
of the FEM by leveraging graph neural networks (GNNs), which are capable of
accounting for the mesh structural information and inductive learning over un-
structured grids and complex topological structures. Empirically, we demonstrate
that the proposed architecture, PhysGNN, promises accurate and fast soft tissue
deformation approximations, and is competitive with the state-of-the-art (SOTA)
algorithms while promising enhanced computational feasibility, therefore suitable
for neurosurgical settings.

## 1   Introduction

Exploiting high-quality preoperative images for surgical planning is a common practice among
neurosurgeons. With the recent advances in surgical tools, medical imagery techniques have especially
made their way into operating rooms to enhance surgical visualization and navigation, giving rise to
image-guided neurosurgical techniques, where correctly aligning patient's intraoperative anatomical
geometry with preoperative images has a significant impact on the accuracy of surgical navigation
[Hagemann et al., 1999, Warfield et al., 2000, Ferrant et al., 2002, Sun et al., 2004, Vigneron et al.,
2004, Choi et al., 2004a, Miller et al., 2007, Wittek et al., 2009, Joldes et al., 2009, Vigneron et al.,
2010, Wittek et al., 2010, Li et al., 2014, Miller et al., 2019]. This task, however, poses a challenge.
During the neurosurgical procedure, the brain continuously undergoes a series of anatomical changes

for a variety of reasons such as gravity, craniotomy,[1] tissue retraction and resection, and anesthetics [Roberts et al., 1998], making surgical navigation based on preoperative data alone error-prone and unreliable [Wittek et al., 2010].

To account for intraoperative brain shift,[2] registration technology, by which the coordinate system of preoperative data and surgical site are brought into spatial alignment, has emerged as an essential tool in image-guided surgical frameworks for maintaining surgical navigation accuracy and depth perception [Crum et al., 2004, Cleary and Peters, 2010, Risholm et al., 2011]. In a registration framework, data alignment is achieved by leveraging interpolation and transformation functions, which manipulate preoperative data to realize tissue deformations and are regularized by a constraint [Ferrant et al., 2002].

At large, registration approaches may be categorized into image-based and model-based methods, which differ in the type of prior information they use as a constraint for deforming preoperative data with respect to intraoperative geometry. To define, image-based registration methods are a set of techniques which mainly use pixel/voxel information and solely rely on image processing techniques to realize tissue deformation [Beauchemin and Barron, 1995, Hata et al., 2000, Ji et al., 2014, Iversen et al., 2018, Tu et al., 2019]. While they may be effective in some settings—e.g., registering deformations of hard tissues such as bone—they are prone to failing in effectively capturing *soft* tissue deformation for various reasons, such as contrast variations between preoperative and intraoperative images resulted from use of different image acquisition techniques, disappearance of boundaries between tissues upon surgical cuts, and presence of contrast agents [Ferrant et al., 2002]. But most importantly, image-based techniques neglect prior information about the biomechanical behaviour of soft tissue, which is an essential knowledge for creating accurate volumetric mappings [Nabavi et al., 2001, Ferrant et al., 2002].

On the other hand, model-based methods are a set of registration techniques that treat images as a *deformable volume*—a notion first introduced by Broit [1981]—to better allow for presenting elastic and plastic deformations. These approaches address intraoperative registration challenges by imposing additional constraints—based on mathematical or physical characterization of soft tissue [Zhang et al., 2017]—to compute soft tissue deformation, resulting in mathematical- and physics-based solution formulations. In particular, studying physics-based methods has become an emerging area of research in recent years [Choi et al., 2004a, Joldes et al., 2009, Malone et al., 2010, Wittek et al., 2010, Joldes et al., 2010, Basafa and Farahmand, 2011, Miller et al., 2019] for their potential to accurately model the biomechanics of the brain and *predicting* soft tissue deformation. Indeed, recent advances in this domain have led to the innovation of advanced image-guided surgical systems that are equipped with augmented reality (AR), which is established by fusing virtual reality (VR) with high-quality preoperative images in the form of a deformable volume [Cleary and Peters, 2010]. The goal of integrating AR in this framework is to facilitate tracking predefined surgical access routes without having to look away from the screen, and provide surgeons with visual tool-tissue interaction [Tonutti et al., 2017].

In the literature, the finite element method (FEM) is a powerful technique for generating 3-dimensional (3D) deformable volumes from high-quality preoperative images, which can simulate intraoperative brain deformations [Wittek et al., 2010]. However, their degree of success boils down to a trade-off between accuracy and speed [Tonutti et al., 2017]. In most cases, benefiting from a finite element (FE) model, which accounts for all types of material and geometric non-linearities, is not always feasible for requiring significant computational resources [Wittek et al., 2010, Tonutti et al., 2017]. To address this limitation, recent works have proposed deriving deformable models by training machine learning algorithms with the results of precomputed FEM simulations that are based on preoperative images [De et al., 2011, Tonutti et al., 2017, Lorente et al., 2017, Pfeiffer et al., 2019, Liu et al., 2020]. However, while promising, they all lack a key aspect for effectively learning over mesh data: they do not account for graph structure—e.g., node connectivities—which is an important piece of information for learning nodal displacements within each FE resulted from prescribed loads. In fact, wanting to incorporate node connectivities by feeding the adjacency matrix as a set of features to the aforementioned algorithms will result in an ill-posed problem as the number of features would then exceed the number of data points. Additionally, these methods are arguably computationally inefficient and hard to train for high quality meshes—which contain many nodes—as the number of

---

[1]Temporarily removing part of the bone from skull to access the brain tissue.

[2]Volumetric brain deformations induced by surgical operations [Nabavi et al., 2001].

parameters needed to train the models would be in order of $O(|\mathcal{V}|)$, where $|\mathcal{V}|$ is the number of mesh nodes. Although the later problem has been previously addressed by training only on a subset of the mesh data [Tonutti et al., 2017], or capturing a lower dimensional embedding of the deformation space through Principle Component Analysis (PCA) [Liu et al., 2020], an important question is raised on whether we can enhance computational feasibility while also benefiting from the structural information of the FE mesh.

**Present Work**    In this work, we propose a novel data-driven framework, named PhysGNN,[3] which approximates soft tissue deformation under prescribed forces by leveraging graph neural networks (GNNs) [Scarselli et al., 2008]. Specifically, PhysGNN is a physics-driven framework for having been trained with the results of Finite Element Analysis (FEA)—that is based on the physical properties of soft tissue such as Young's modulus and Poisson ratio—and for approximating tissue deformation— i.e., displacement of the FE mesh nodes—by summing up the forces applied to the prescribed load nodes and their neighbours using GraphSAGE [Hamilton et al., 2017] and GraphConv [Morris et al., 2019] layers, analogous to the way displacement caused by applied force(s) is computed according to the Newton's first law of motion. To the best of our knowledge, we are the first ones to be leveraging GNNs for approximating soft tissue behavior upon prescribed loads, and propose combination of GraphSAGE and GraphConv layers as well as Jumping Knowledge (JK) connections [Xu et al., 2018] for this purpose. Our motivation for using GNNs for this task is their ability to enable generalization of convolution on graph data through the notion of *neural message passing* [Gilmer et al., 2017], by which the information of nodes within a neighbourhood of an arbitrary size are transformed and aggregated producing a new value [Hamilton, 2020]. The message passing framework enables inductive learning over graphs through *parameter sharing*, a property that ensures learning a constant number of parameters independent of the mesh size. Therefore, contrary to the previous methods, not only is PhysGNN capable of incorporating mesh structural information—i.e., proximity of mesh nodes with each other and the Euclidean distance of neighbouring nodes from one another—which can aid with effective prediction of tissue deformation, but also can enhance computational feasibility when learning from high quality meshes that consist of many nodes.

We evaluate the performance of various PhysGNN models—resulted from different combination of GraphSAGE and GraphConv layers and aggregation functions—on two datasets—distinguished by the number of force load points and the total force applied—against the results of FEA (our baseline). Empirically, we demonstrate that PhysGNN is capable of successfully predicting tissue deformation upon prescibed forces under 0.005 and 1.9 seconds using a GPU and CPU, respectively. In fact, the absolute displacement error of the best-performing PhysGNN model for 94% to 97% of the nodes was found to be under 1 mm—which is the precision in neurosurgery [Miller et al., 2010].

## 2   Related Work

**Approximating Tissue Deformation with Machine Learning**    The predictive power of various machine learning algorithms in approximating tissue deformation with respect to physics laws has been assessed in a number of studies. These approaches either implement machine learning techniques as an alternative to the FEM—e.g., representing mass points as cellular neural networks to assimilate elastic deformations [Zhong et al., 2006, Zhang et al., 2019]—or as a subsequent step to preoperative FEA in which data-driven models are resulted by training machine learning models with patient-generated content that are intrasubject (patient-specific) to predict soft tissue deformation, as shown in Figure 1. For example, while De et al. [2011] proposed a physics-driven framework, named PhyNNeSS, consisting of radial basis function to reconstruct the deformation fields of liver and Penrose drain models, Lorente et al. [2017] investigated the success of linear regression, decision trees, random forests, and extremely randomized trees in simulating the biomechanical behavior of human liver under large deformations and during breathing in real time. Furthermore, Liu et al. [2020] exploited ANNs to reconstruct tissue deformation of a head and neck tumour, where the dataset was encoded into a low-dimensional representation using PCA prior to training to facilitate the learning process. Meanwhile, similar techniques have been employed to approximate tissue deformation in neurosurgical settings. In [Tonutti et al., 2017], the authors explored and compared the effectiveness of ANNs and support vector regression (SVR) in predicting deformation of a brain tumour caused by applying forces to the brain tissue. Generalizing the notion of deep learning based soft tissue

---

[3]The data and implementation are available at `https://github.com/YasminSalehi/PhysGNN`.

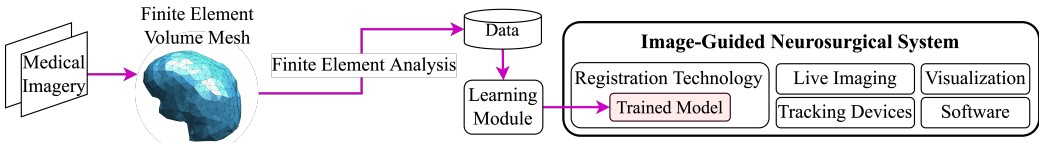

Figure 1: A generic image-guided neurosurgical system framework [Cleary and Peters, 2010], where patient's anatomy is aligned with preoperative images through deforming a deformable volume generated from preoperative imagery, which its biomechanical behavior under prescribed loads is predicted by machine learning methods.

simulation frameworks to different types of organs, Pfeiffer et al. [2019] proposed a convolutional neural network (CNN) based soft tissue simulation system to predict an organ's internal deformation from known surface deformation. In their study, they accounted for *all* tissue types within an organ and trained CNNs on an entirely synthetic data of random organ-like meshes to enhance the learning process by using more data than otherwise available. They stated their method can be adopted by the neurosurgical community to capture brain shift, or be used in motion compensation radio-therapy.

**Learning Mesh-Based Simulations Using GNNs**   Recently, Sanchez-Gonzalez et al. [2020] and Pfaff et al. [2020] have proposed Graph Network-based Simulators (GNS) and MeshGraphNets, respectively, to simulate interaction of fluids, rigid solids, and deformable materials with one another [Sanchez-Gonzalez et al., 2020], and complex physical systems such as aerodynamics, structural mechanics, and cloth [Pfaff et al., 2020]. Both GNS and MeshGraphNets are generative architectures that use GraphNet [Sanchez-Gonzalez et al., 2018] as their building block.

## 3 The Physics-Driven Graph Neural Network Based (PhysGNN) Model

PhysGNN is a data-driven model that leverages GNNs to learn the nonlinear behavior of soft tissue under prescribed loads from FE simulations. In this work, GNNs are deemed as the most suitable deep learning candidates for capturing tissue deformation for the reasons stated below.

1. First, GNNs enable incorporating mesh structural information by forming a computation graph—a neural network architecture—for every node through unfolding the neighbourhood around them via neural message passing, which refers to the exchange of feature and structural information between nodes that are within the same $k$-hop neighbourhood [Hamilton, 2020]. This characteristic of GNNs can ensure more accurate prediction of tissue deformation that is especially in close proximity of prescribed forces and susceptible to undergoing larger displacements, since the distance between prescribed force nodes and free-boundary condition nodes can be accounted for.

2. Second, GNNs learn the same weight and bias parameters across the FE mesh through parameter sharing. In other words, the number of parameters which need to be learned are constant and independent of the mesh size, a property that also enables generalization of GNNs to unseen parts of the graph. This makes training PhysGNN with high quality meshes—which inherently consist of many nodes—computationally less expensive than the methods presented in the previous studies.

### 3.1 Architecture

PhysGNN is a data-driven model that captures tissue deformation—i.e., displacement of the FE mesh nodes—caused by prescribed forces microscopically by summing up or taking the maximum of forces applied to their $k$-hop neighbours at each $k$ for $K$ iterations, where $k \in \{1, ..., K\}$. The idea of force summation—or taking the maximum for when force is applied to only one of the mesh nodes—is based on the way net displacement due to a force is computed according to Newton's first law of motion, and inspired by the force propagation model presented in [Choi et al., 2004b]. For this purpose, PhysGNN incorporates two types of GNNs, namely GraphSAGE [Hamilton et al., 2017] and GraphConv—the graph convolution operator of $k$-GNNs in [Morris et al., 2019]—where the later GNN allows for incorporating nodal distances. The message passing architecture of GraphSAGE

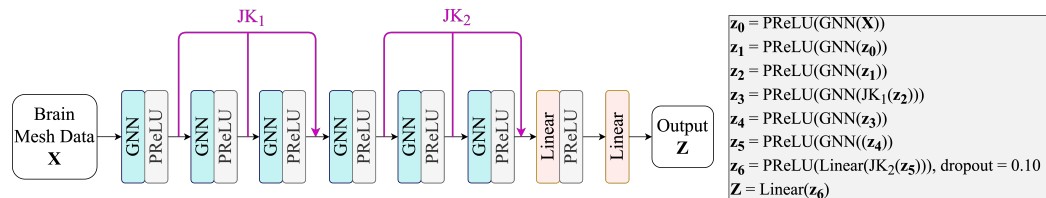

Figure 2: Architectural diagram of PhysGNN where the "GNN" layers may be GraphConv and/or GraphSAGE layers that aggregate messages by summation and/or maximum functions. The best-performing PhysGNN models in our study are shown in Figure 4a and 4b.

and GraphConv for a graph $\mathcal{G} = (\mathcal{V}, \mathcal{E})$, with $\mathcal{V}$ and $\mathcal{E}$ being the vertex and edge sets, respectively, $u, v \in \mathcal{V}$, and matrix $\boldsymbol{X} \in \mathbb{R}^{d \times |\mathcal{V}|}$ representing the node features, are defined as:

$$
\mathbf{m}_{\mathcal{N}(u)}^{(k)} = \text{AGGREGATE}^{(k)}(\{\mathbf{h}_v^{(k-1)}, \forall v \in \mathcal{N}(u)\})
$$
$$
\mathbf{h}_u^{(k)} = \sigma\left(\mathbf{W}_u^{(k)} \cdot \mathbf{h}_u^{(k-1)} \oplus \mathbf{W}_{\mathcal{N}(u)}^{(k)} \cdot \mathbf{m}_{\mathcal{N}(u)}^{(k)} + \mathbf{b}^{(k)}\right), \tag{1}
$$

and

$$
\mathbf{m}_{\mathcal{N}(u)}^{(k)} = \text{AGGREGATE}^{(k)}(\{e_{vu} \cdot \mathbf{h}_v^{(k-1)}, \forall v \in \mathcal{N}(u)\})
$$
$$
\mathbf{h}_u^{(k)} = \sigma\left(\text{MERGE}^{(k)}\left(\mathbf{W}_u^{(k)} \cdot \mathbf{h}_u^{(k-1)}, \mathbf{W}_{\mathcal{N}(u)}^{(k)} \cdot \mathbf{m}_{\mathcal{N}(u)}^{(k)}\right) + \mathbf{b}^{(k)}\right), \tag{2}
$$

respectively, where $\mathcal{N}(u)$ is the neighbourhood of node $u$, $\mathbf{m}_{\mathcal{N}(u)}^{(k)}$ represents the aggregated messages from the neighbours of node $u$ at iteration $k$, where $k \in \{1, \ldots, K\}$, $\mathbf{h}_u$ and $\mathbf{h}_v$ are the embedding for nodes $u$ and $v$, respectively, $e_{uv}$ is the edge weight between node $u$ and $v$, $\oplus$ is the concatenation operator, AGGREGATE is an arbitrary permutation-invariant, differentiable function, MERGE is either a sum, a column-wise vector concatenation or a long short-term memory (LSTM) style update step, $\sigma$ is a component-wise non-linear function (e.g., a sigmoid, or a ReLU), and the weights, $\mathbf{W}_u^{(k)}, \mathbf{W}_{\mathcal{N}(u)}^{(k)} \in \mathbb{R}^{d^k \times d^{(k-1)}}$, and the bias term, $\mathbf{b}^{(k)} \in \mathbb{R}^{d^{(k)}}$, are trainable parameters. Moreover, the input to the GNN is simply the node features, expressed as $\mathbf{h}_u^0 = \mathbf{x}_u, \forall u \in \mathcal{V}$, while the output of the final layer for each node is $\mathbf{z}_u = \mathbf{h}_u^K, \forall u \in \mathcal{V}$ [Hamilton et al., 2017, Morris et al., 2019, Hamilton, 2020].

In this work, $K$ is set to 6—i.e., the mesh nodes capture information from their 6-hop neighborhood. This decision was made by noting performance improvement after increasing the number of GNN layers. Although a deeper neural network architecture may have yielded a better performance, increasing the number of GNN layers beyond 6 increased training time while may have also resulted in over-smoothing for aggregating over the entire FE mesh. Moreover, the implemented AGGREGATE functions are addition and maximum for summing up or taking the maximum of the applied forces, respectively, while the MERGE function used for the GraphConv layers is concatenation. PhysGNN also benefits from a variation of JK connections [Xu et al., 2018] to combat over-smoothing, which generates a weighted sum of embeddings at different iterations from a bi-directional LSTM layer as:

$$
\sum_{t=1}^{T} \alpha_u^{(t)} \mathbf{h}_u^{(t)}, \tag{3}
$$

where $\alpha_u^{(t)}$ are the attention scores. A full architectural diagram of PhysGNN is depicted in Figure 2.

It should be noted that different PhysGNN models can be obtained by varying the combination of GraphConv and GraphSAGE layers and aggregation functions. The performance of various PhysGNN models on two different datasets is presented in the Results and Discussion section.

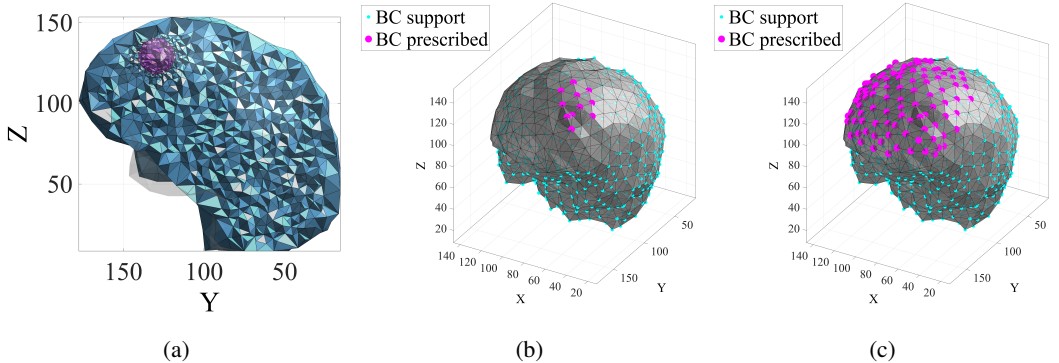

Figure 3: (a) Cut view of the FE volume mesh, where cyan represents the healthy brain tissue, while magenta represents the tumour tissue; prescribed and fixed boundary condition nodes, denoted as BC prescribed and BC support, respectively, of (b) Dataset 1 and (c) Dataset 2.

Table 1: Material properties of different parts of the brain [Joldes et al., 2010, Wittek et al., 2010].

| Dataset | Density (kg/m$^3$) | Young's Modulus (Pa) | Poisson Ratio |
|---|---|---|---|
| Healthy Tissue | 1000 | 3000 | 0.49 |
| Tumour Tissue | 1000 | 7500 | 0.49 |

## 4  Experimental Setting

### 4.1  Data Acquisition

**Generating the Finite Element Mesh**   The FE volume mesh for running FE simulations was generated by running TetGen [Si, 2015] on the surface mesh files provided by Tonutti et al. [2017], which is based on MRI scans taken from the Repository of Molecular Brain Neoplasia Data [Madhavan et al., 2009]. Figure 3a is illustrative of the FE volume mesh, which consists of 9118 nodes. More details on the FE volume mesh may be found in the supplementary material.

**Material Properties**   Based on the work done in [Joldes et al., 2010] and [Wittek et al., 2010], the mechanical behavior of healthy brain and tumour tissues were characterized as hyper-elastic material obeying the Neo-Hookean constitutive law [Maas et al., 2012]. Additionally, similar to the experimental setting in [Tonutti et al., 2017], the tumour and healthy brain tissues have identical density and Poisson ratio, and the tumour is assumed to be stiffer than the healthy brain tissue, thus having a larger Young's modulus. The numerical value of material parameters are listed in Table 1.

**Finite Element Simulations**   To assess the effectiveness of PhysGNN in predicting brain shift imposed by forces during surgery and especially craniotomy—which is modeled by applying forces to the surface of the brain [Wittek et al., 2009, Miller et al., 2019]—two different datasets were generated by running FE simulations on the FE mesh in FEBio [Maas et al., 2012] using the GIBBON [Moerman, 2018] toolbox. The datasets are distinguished by the total force applied and the number of prescribed load nodes. Specifically, Dataset 1 was created by applying 1.35 N to 1 node at a time in 30 time steps and 15 directions for a total of 11 different nodes. This choice was motivated by the results of the experimental study done by Gan et al. [2015] who measured 70% of induced forces during neurosurgery to be less than 0.3 N and a maximal peak at 1.35 N during tissue dissection. On the other hand, Dataset 2 was created by applying 20 N to 100 nodes at a time in 30 time steps and 165 directions to further capture the non-linear behavior of soft tissue under large forces. Therefore, the amount of force applied to each prescribed node in Dataset 1 at time $i$ is:

$$\mathbf{F}_i = \frac{\mathbf{F}_{\text{total}}}{30} \times i, \ i \in \{1, \ldots, 30\}, \tag{4}$$

Table 2: Characteristics of Datasets 1 and 2. B.C. stands for Boundary Condition. Pres. stands for Prescribed.

| Dataset | No. Fixed B.C. Nodes | No. Pres. Load Nodes | No. Pres. Load Node(s) per Simulation | No. Force Magnitudes | Max. Force Applied (N) | No. Directions | No. Simulations |
|---------|------|------|------|------|------|------|------|
| 1 | 502 | 11 | 1 | 30 | 1.35 | 15 | 4950 |
| 2 | 502 | 100 | 100 | 30 | 20 | 165 | 4950 |

Table 3: PhysGNN features and outputs.

| Features ($\mathbf{X}$) | Output ($\mathbf{Z}$) |
|---|---|
| $F_x, F_y, F_z, F_\rho, F_\theta, F_\phi$, Physical Property | $\delta x, \delta y, \delta z$ |

whereas the amount of force applied to one of the prescribed nodes in Dataset 2 at time $i$ is:

$$\mathbf{F}_i = \frac{\mathbf{F}_{\text{total}}}{30 \times 100} \times i, \ i \in \{1, \dots, 30\}. \tag{5}$$

The direction of forces applied to each prescribe load node features its surface normal direction, as well as directions that are randomly sampled from a hemisphere with radius 1. Lastly, to train PhysGNN, a train:validation:test split of 70:20:10 was applied to both datasets. Figures 3b and 3c are illustrative of the prescribed load nodes in Datasets 1 and 2, respectively, which are located near the tumour site, as well as the fixed boundary condition nodes, which mimic the skull bone that prevents tissue from moving. Further details on the generated datasets can be found in Table 2.

## 4.2 Features and Output

The features inputted into PhysGNN are the value of prescribed forces applied to the mesh nodes in Cartesian and spherical coordinates, denoted as $(F_x, F_y, F_z)$ and $(F_\rho, F_\theta, F_\phi)$, respectively, as well as a constant value, named Physical Property, which for nodes with free boundary condition is set to 0.4 if the node belongs to tumour tissue and 1 if the node belongs to healthy brain tissue, and 0 for nodes with fixed boundary condition. This value determines how much a node can undergo displacement based on its boundary condition and Young's modulus given that brain tissue is susceptible to undergoing larger displacements for its smaller Young's modulus compared to the tumour tissue. In particular, the value of 0.4 was calculated as the ratio of healthy tissue's Young's modulus (3000 Pa) over tumour tissue's Young's modulus (7500 Pa). On the other hand, the output of PhysGNN are the displacement value of the mesh nodes in the $x, y$, and $z$ directions, denoted as $\delta x, \delta y, \delta z$. Lastly, for the GraphConv models, the edge weights, $e_{u,v}$, were calculated as the inverse of the Euclidean distance between adjacent nodes $u$ and $v$ with $u, v \in \mathcal{V}$ as:

$$e_{u,v} = \frac{1}{\sqrt{((x_u - x_v)^2 + (y_u - y_v)^2 + (z_u - z_v)^2)}}. \tag{6}$$

A summary of the features and output of PhysGNN can be found in Table 3.

## 4.3 Baseline

Similar to the previous studies [Tonutti et al., 2017, Lorente et al., 2017, Liu et al., 2020], the predictions made by the proposed data-driven model, namely PhysGNN in this work, are compared against the results approximated by the FEM.

## 4.4 Loss Function, Optimization, and Regularization

The loss function used for learning the trainable parameters is the mean Euclidean error computed as:

$$\mathcal{L} = \frac{1}{|\mathcal{V}|} \sum_{v \in \mathcal{V}} \sqrt{\sum_{i=1}^{3} (\mathbf{y}_{v,i} - \mathbf{z}_{v,i})^2}, \tag{7}$$

Table 4: Best performance of PhysGNN on Datasets 1 and 2.

| Dataset | MAE ($\delta x$) (mm) | MAE ($\delta y$) (mm) | MAE ($\delta z$) (mm) | Mean Euclidean Error (mm) | Euclidean Error $\leq$ 1 mm (%) | Mean Absolute Position Error (mm) | Absolute Position Error$\leq$ 1 mm (%) |
|---|---|---|---|---|---|---|---|
| Dataset 1 Validation | 0.1145 $\pm$ 0.2832 | 0.1051 $\pm$ 0.2544 | 0.1009 $\pm$ 0.2650 | 0.2105 $\pm$ 0.4530 | 94.99 | 0.1690 $\pm$ 0.4200 | 95.91 |
| Dataset 1 Test | 0.1117 $\pm$ 0.2750 | 0.0999 $\pm$ 0.2318 | 0.0999 $\pm$ 0.2601 | 0.2049 $\pm$ 0.4329 | 95.11 | 0.1612 $\pm$ 0.3961 | 96.18 |
| Dataset 2 Validation | 0.1478 $\pm$ 0.3825 | 0.1879 $\pm$ 0.5527 | 0.1283 $\pm$ 0.2364 | 0.3116 $\pm$ 0.6958 | 94.25 | 0.2106 $\pm$ 0.6129 | 96.26 |
| Dataset 2 Test | 0.1393 $\pm$ 0.3402 | 0.1842 $\pm$ 0.5479 | 0.1249 $\pm$ 0.2272 | 0.3023 $\pm$ 0.6671 | 94.60 | 0.2023 $\pm$ 0.5920 | 96.59 |

Table 5: The test set statistics of Datasets 1 and 2, where $\mathbf{y}$ is the displacement, and Max. Euclidean Error$_{\text{mean}}$ is computed as the average of the maximum Euclidean error observed for each data element—i.e., each simulation.

| Dataset | $\delta \mathbf{y}_{\text{max}}$ (mm) | $\delta \mathbf{y}_{\text{mean}}$ (mm) | Max. Euclidean Error$_{\text{mean}}$ (mm) |
|---|---|---|---|
| 1 | 24.5864 | 11.6229 $\pm$ 5.7681 | 2.5924 $\pm$ 1.5665 |
| 2 | 47.8233 | 16.4170 $\pm$ 10.4650 | 4.1952 $\pm$ 2.5161 |

where $|\mathcal{V}|$ is the number of mesh nodes, $\mathbf{y} \in \mathbb{R}^{|\mathcal{V}| \times 3}$ is the displacement in the $x$, $y$, and $z$ directions approximated by the FEM, and $\mathbf{z} \in \mathbb{R}^{|\mathcal{V}| \times 3}$ is the displacement predicted by PhysGNN. To minimize the loss value, AdamW optimizer was used with an initial learning rate of 0.005, which was reduced by a factor of 0.1 to a minimum value of $1e-8$ if no improvement on the validation loss was noticed after 5 epochs. To prevent over-fitting, early stopping was one of the regularization methods implemented by which learning stopped after no decrease in validation loss occurred after 15 epochs. Moreover, a dropout rate of 0.1 was applied to the second last layer of PhysGNN to improve generalization to unseen data.

## 5 Results and Discussion

### 5.1 Performance of PhysGNN in Predicting Tissue Deformation

Table 4 is indicative of the performance of the PhysGNN models that resulted in the lowest validation loss on Datasets 1 and 2, illustrated in Figures 4a and 4b, respectively. According to the results, PhysGNN is capable of effectively predicting tissue deformation under prescribed loads, especially highlighted by 96% and 97% of absolute position errors being under 1 mm in Datasets 1 and 2, respectively. Notably, approximating tissue deformation using the FEM on a quad-core Intel i7 @ 2.9 GHz CPU took $11.5099 \pm 6.8004$ seconds on average for each FE simulation, with minimum and maximum computation times of $5.4191$ and $32.3798$ seconds, respectively, whereas prediction of tissue deformation per each FE simulation on a dual-core Intel(R) Xeon(R) @ 2.20GHz CPU took $1.8294 \pm 0.0322$ seconds on average, while on an NVIDIA P-100 GPU took $0.0043 \pm 0.0005$ seconds on average. Based on these observations and the fact that the accuracy in neurosurgery is not better than 1 mm [Miller et al., 2010], PhysGNN manifests itself as a promising deep learning module for predicting tissue deformation in neurosurgical settings for its accurate approximations, computation speed, and ease of implementation. Additional results on the median of errors may be found in the supplementary material.

### 5.2 Comparison to Similar Studies

Comparing our results with the most recent works in predicting tissue deformation with machine learning methods, our approach compares favourably. In [Tonutti et al., 2017] the authors investigated

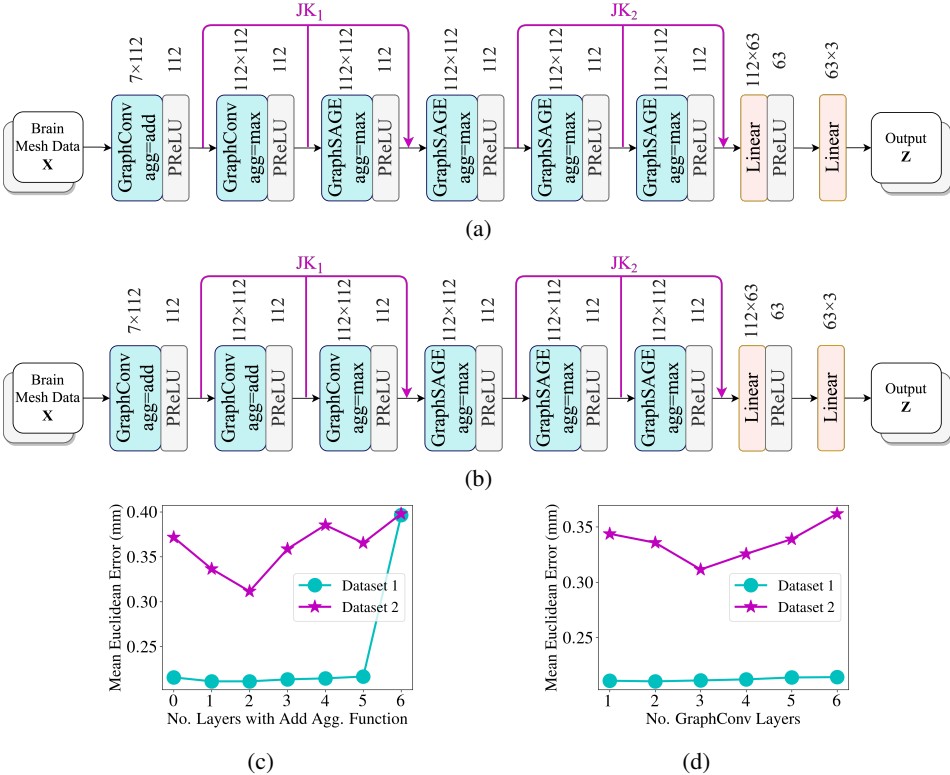

Figure 4: Best-performing PhysGNN models on (a) Dataset 1, and (b) Dataset 2. The effect of (c) consecutively replacing the maximum with the addition aggregation function in a PhysGNN model consisting of 3 GraphConv layers followed by 3 GraphSAGE layers, and (d) consecutively replacing GraphSAGE with GraphConv layers while using the best-performing aggregation function combination determined from Figure 4c. Agg. stands for aggregation.

the effectiveness of their approach in approximating a brain tumour tissue deformation caused by applying forces less than 1 N to a single node on the brain surface and reported a mean absolute position error of $0.191 \pm 0.201$ mm. While our absolute position error for Dataset 1 (where forces $\leq$ 1.35 N are applied to a single node) seems to only be marginally smaller than theirs, we argue that our result is implicitly much better as our test set contains larger deformations since we consider for both the brain and tumour tissues. Similarly, our proposed method appears to be competitive with the results in [Lorente et al., 2017] where they reported 100% of the Euclidean errors to be $\leq$ 1 mm, for a dataset with a maximum displacement of 15 mm. Lastly, Liu et al. [2020] reported an average Euclidean error of 0.129 mm and 0.392 mm, and an average maximum Euclidean error of 0.483 mm and 1.011 mm for two different test sets that exhibited a maximum displacement of 30 mm on a mesh with 1158 nodes. By referring to Tables 4 and 5, our results (achieved on a mesh with 9118 nodes) are shown to be competitive without requiring PCA to reduce the subspace dimensionality. Indeed, GNNs are inherently graph embedding models that encode graphs into a lower dimensional subspace to facilitate data manipulation for carrying out machine learning tasks on graphs. Therefore, we deem our method as not only effective for predicting tissue deformation but also a simpler one to implement.

## 5.3 Ablation Study

To find the best-performing PhysGNN model on different datasets, we assessed the performance of various PhysGNN models resulted from different combinations of GNN layers and aggregation functions. First, to find the best-performing aggregation function combination, we started with a PhysGNN model consisting of three GraphConv layers followed by three GraphSAGE layers with all six layers using the maximum aggregation function, which we consecutively replaced with the addition aggregation function. Afterwards, we initialized a PhysGNN model with one

GraphConv layer followed by five GraphSAGE layers and replaced GraphSAGE with GraphConv layers consecutively while using the best-performing combination of aggregation functions.

Initializing the PhysGNN model with three GraphConv layers followed by three GraphSAGE layers, consecutive replacement of maximum with addition aggregation function for two iterations reduced the mean validation Euclidean loss of Dataset 2 more noticeably, while such replacement for five iterations only marginally changed the performance of PhysGNN on the validation set of Dataset 1, as shown in Figure 4c. Similarly, consecutive replacement of GraphSAGE with GraphConv layers while using the best-performing combination of aggregation functions resulted in a more pronounced decrease in the validation loss of Dataset 2—where using three GraphConv layers yielded the best performance— while caused over-smoothing beyond a certain number for both datasets, as depicted in Figure 4d.

These observations are justified by realizing that nodal displacements in Dataset 1 are resulted from applying forces to only one node per simulation, contrary to nodal displacements in Dataset 2, which occur from multiple prescribed force nodes per simulation. Therefore, for Dataset 2, not only is using the summation aggregation function a more reasonable choice, but also weighting the impact of multiple force load points on their neighboring nodes by multiplying their message with the inverse of the nodal distance is indeed expected to produce more accurate results. The later observation sheds light on the importance of incorporating nodal distance and proper formulation of edge weights to effectively capture tissue deformation.

It is worthy to note that the best-performing PhysGNN model on Dataset 2, depicted in Figure 4b, resulted in a validation and test mean Euclidean error of $0.2114$ mm and $0.2058$ mm, respectively, for Dataset 1, which is only marginally greater than the errors resulted from the best-performing model for Dataset 1, depicted in Figure 4a. Accordingly, the PhysGNN model depicted in Figure 4b may be regarded as robust for both datasets.

### 5.4 Limitations and Future Directions

One of the limitations of this study is that comparing our method against other works was done on empirical grounds due to either lack of accessibility to other studies' data and/or implementation, or not having sufficient computational power to use their data/models. However, our work presents itself as a great benchmark for future studies as our data and implementations are publicly accessible and reproducing our results only requires a Google Colab Pro server.

Another limitation of this work is using the FEA results as baseline, which is dependent on the quality of MRI image segmentation, and the selected soft tissue constitutive laws that may not hold true for different patients [Tonutti et al., 2017]. Moreover, tetrahedral elements are prone to undergoing volumetric locking for almost incompressible materials, such as soft tissue, and should not be used to simulate tissue deformation [Joldes et al., 2019]. To alleviate these problems, Meshless Total Lagrangian Explicit Dynamics (MTLED) algorithms are emerging as a solution to better simulate soft tissue deformation, which can also be trained with empirical results [Miller et al., 2019, Joldes et al., 2019]. As PhysGNN may also be trained with the results of Lagrangian meshless methods, the presented framework is foreseen to yield highly accurate and computationally efficient data-driven deformable volumes in the future.

## 6 Conclusion

The purpose of this work was to introduce a novel data-driven framework, PhysGNN, to combat the limitations of using the FEM for approximating tissue deformation—and especially craniotomy-induced brain shift—in neurosurgical settings. Through our empirical results, we demonstrated the effectiveness of PhysGNN in accurately predicting tissue deformation. Compared to similar studies, our method is competitive with data-driven SOTA approaches for capturing tissue deformation while promising enhanced computational feasibility, especially for large and high-quality meshes.

### Acknowledgments and Disclosure of Funding

The authors would like to thank the reviewers for their helpful and constructive feedback. This work was supported by the Natural Sciences and Engineering Research Council (NSERC) of Canada.

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
