# Supplemental Material for PhysGNN: A Physics–Driven Graph Neural Network Based Model for Predicting Soft Tissue Deformation in Image–Guided Neurosurgery

**Yasmin Salehi, Dennis Giannacopoulos**
Department of Electrical and Computer Engineering
McGill University
Montreal, QC, Canada
yasmin.salehi@mail.mcgill.ca, dennis.giannacopoulos@mcgill.ca

## A  Finite Element Volume Mesh

Table 1 provides information on the finite element (FE) volume mesh that was used for generating Datasets 1 and 2.

Table 1: Finite element volume mesh statistics.

| Attribute | Value |
|---|---|
| Mesh points | 9118 |
| Mesh tetrahedra | 55927 |
| Mesh faces | 112524 |
| Mesh faces on exterior boundary | 1340 |
| Mesh faces on input facets | 2992 |
| Mesh edges on input segments | 4488 |
| Steiner points inside domain | 7618 |

## B  Infrastructure Settings

The FE simulations in our study were carried out on quad-core Intel i7 @ 2.9 GHz CPU, while different PhysGNN models were trained on a Google Colab Pro server with 23GB of RAM and one NVIDIA P-100 GPU with 16 GB of video RAM.

## C  Additional Results

The table below represents the median of absolute error in the $x$, $y$, and $z$ directions denoted as $\delta x$, $\delta y$, and $\delta z$ respectively, Euclidean error, and absolute position error.

36th Conference on Neural Information Processing Systems (NeurIPS 2022).

Table 2: Median of the evaluation metrics resulted from the best-performing PhysGNN models on Datasets 1 and 2.

| Dataset | Median Absolute Error ($\delta x$) (mm) | Median Absolute Error ($\delta y$) (mm) | Median Absolute Error ($\delta z$) (mm) | Median Euclidean Error (mm) | Median Absolute Position Error (mm) |
|---|---|---|---|---|---|
| Dataset 1 Validation | 0.0199 | 0.0228 | 0.0221 | 0.0531 | 0.0303 |
| Dataset 1 Test | 0.0200 | 0.0227 | 0.0224 | 0.0537 | 0.0303 |
| Dataset 2 Validation | 0.0436 | 0.0537 | 0.0538 | 0.1295 | 0.0656 |
| Dataset 2 Test | 0.0426 | 0.0534 | 0.0529 | 0.1285 | 0.0646 |

## D   Comparison to Similar Studies

The summary table below compares our results with a few similar studies based on empirical grounds.

Table 3: Comparing our method with other studies on empirical basis.

| Study | Number of Nodes in the FE Mesh | Maximum Displacement in the Dataset(s) (mm) | Mean Absolute Position Error (mm) | Mean Euclidean Error (mm) | % of Euclidean Error Below 1 mm | Average of Maximum Euclidean Error per Simulation (mm) |
|---|---|---|---|---|---|---|
| Tonutti et al. [2017] | 1087 | — | 0.191 | 0.18 | — | — |
| Lorente et al. [2017] | 318960 – 494310 | 15 | — | 0.07 | 100 | — |
| Liu et al. [2020] | 1158 | 30 | — | 0.129 | 98 | 0.483 |
| | 1158 | 30 | — | 0.392 | 98 | 1.011 |
| **PhysGNN (ours)** | 9118 | 24.5864 | 0.1612 | 0.2049 | 95.11 | 2.5924 |
| | 9118 | 47.8233 | 0.2023 | 0.3023 | 94.60 | 4.1952 |