# OpenReview forum: "PhysGNN: A Physics--Driven Graph Neural Network Based Model for Predicting Soft Tissue Deformation in Image--Guided Neurosurgery"
_NeurIPS.cc/2022/Conference — NeurIPS 2022 Accept_

### Official Review · Reviewer_PQLo · 2022-06-23

**Rating:** 7
**Confidence:** 3
**Soundness:** 3 good
**Presentation:** 3 good
**Contribution:** 3 good

**Summary:**

In this paper, the authors present the creation of a surrogate FEM model of the biomechanical behavior of the brain. The creation of this surrogated model is based on the use of Graph Neural Network (GNN) for its ability to allow introducing not only the positions of the nodes but also the connections between them (by sending messages). Compared to similar approaches, the model is competitive in terms of speed and accuracy. In addition, the authors state that their approach allows the generation of models surrogated to different approximations by means of finite elements.

**Questions:**

*.- How could this approximation be applied in real scenarios?
*.- Could it be applied to the simulation of anisotropic tissue behavior?
*.- The images are a little bit poor. They should be improved.


**Limitations:**

Authors have adequately addressed the limitations of their approach.

**Strengths And Weaknesses:**

Strengths:
*.- This approximations allow to generate fast and accurate surrogate models to FEM based on GNN.
*.- Due to the use of GNN, it can consider connexions between nodes.
*.- It is general enough to be applied to other Finite Element approximations.

Weaknesses:
*.- In this approximation (as other ones) only forces are considerer when (f.e. in case of brain-shift) most of the time only the displacement of the nodes can be obtained.
*.- It is only compared to synthetic models.

---

> ### Author Response · Authors · 2022-08-02
> **Response to Reviewer PQLo**
>
> We would like to thank you for your time and positive review regarding the quality of our paper and its contributions. We now answer the questions you have raised in your review below.
>
> **Application of PhysGNN in Real Scenarios**
>
> Our framework captures tissue deformation by taking the amount of force applied to the tissue as its input. Therefore, to compute tissue deformation in real scenarios, the forces applied to soft tissue need to be quantified in some way. In the studies done by Gan et al., [2015] and Maddahi et al., [2016] force-sensing bipolar forceps (alongside other force sensing components in the later study) are used to measure forces in neurosurgical procedures. Consequently, similar tools can be used to capture the forces exerted to the brain tissue which can then be fed into PhysGNN to approximate tissue deformation and update the preoperative data—e.g. the finite element mesh constructed from preoperative high quality MRI images—to have it aligned with intraoperative geometry.
>
> **Application of PhysGNN for Capturing Anisotropic Tissue Behaviour Under Prescribed Forces**
>
> We foresee our method to be applicable for capturing deformation for anisotropic tissue. Since anisotropic tissue’s behavior under prescribed loads is dependent on the direction at which the force is being applied to it, it would be beneficial to add the physical properties (e.g. Young’s modulus, shear modulus) of the material in different directions to the feature set of the mesh nodes. Moreover, the dataset should include a sufficient number of training examples for all directions at which the force could be applied to it.
>
> **Possible Points for Improvement**
>
> We agree that the quality of the images in our paper can be improved. We will definitely improve and update them in the future revision of the paper.
>
> **Thank You**
>
> We hope we have addressed your questions and concerns and are happy to further discuss with you during the author-reviewer discussion period.
>
> **References**
>
> Liu Shi Gan, Kourosh Zareinia, Sanju Lama, Yaser Maddahi, Fang Wei Yang, and Garnette R Suther- land. Quantification of forces during a neurosurgical procedure: a pilot study. World neurosurgery, 84(2):537–548, 2015.
>
> Yaser Maddahi, Jordan Huang, Jade Huang, Liu Shi Gan, Hamidreza Hoshyarmanesh, Kourosh Zareinia, and Garnette R Sutherland. Real-time measurement of tool-tissue interaction forces in neurosurgery: Quantification and analysis. In 2016 IEEE international conference on advanced intelligent mechatronics (AIM), pages 1405–1410. IEEE, 2016.

---

### Official Review · Reviewer_8xpx · 2022-07-11

**Rating:** 6
**Confidence:** 4
**Soundness:** 3 good
**Presentation:** 3 good
**Contribution:** 3 good

**Summary:**

In this paper, the authors present a method for soft tissue deformation of the brain using GNNs. The GNN is trained with the results of a FEM simulation. Compared to other methods that were trained on FEM results, the authors use a GNN to account for the graph structure of the FEM. The results show that the presented method is on par with other methods for soft tissue deformation of the brain.

**Questions:**

1) Is there a specific reason to use two different aggregation functions (mean and max)?
2) Are the node features that are used as input the same for all nodes?
3) Why is the input provided in polar and Cartesian coordinates?
4) Is the output of the last linear layer 63x3?


**Limitations:**

Yes, the authors have addressed the limitations of their work.

**Strengths And Weaknesses:**

The paper is very well written and organized, especially the introduction and the prior work section is very detailed. The method part is missing some details and is difficult to understand. For example, the description (177 – 186) of Eq. 1 and 2. Having a detailed description of each element of Eq. 1 and 2 would improve the understanding. The evaluation could be more detailed as only the mean and std error is provided. A more detailed evaluation of the different regions of the brain would be beneficial.

---

> ### Author Response · Authors · 2022-08-02
> **Response to Reviewer 8xpx**
>
> We would like to thank you for your time and positive comments on our work. We now address the questions and concerns you have raised in your review.
>
> **Use of Summation and Maximum Aggregation Functions**
>
> As mentioned in lines 172-174 of our paper, the intuition behind using the summation aggregation function is to add up the forces applied to a node and its k-hop neighbors for means of approximating tissue deformation similar to how we compute the net displacement caused by the net force using Newton’s first law of motion. In scenarios where force is applied to only one node or one of the k-hop neighbors, the net force is essentially the maximum force applied. By performing an ablation study, where we consecutively replaced the maximum aggregation function with the summation aggregation function, we noticed that using the summation aggregation function for the first two GNN layers effectively decreased the mean Euclidean error on dataset 2 (where force is applied to multiple nodes at once) while it did not change it very much on dataset 1 (where force is applied to only one node at a time) until the 6th layer, as shown in Figure 4c, which was in line with our expectations.
>
> **Node Features and Output**
>
> The node features are the same for all nodes, which are the force applied to each node in cartesian and spherical coordinates, as well as a value named Physical Property, which for nodes with free boundary condition takes a value of
>
> * 0.4 if the node belongs to the tumor tissue, and
>  * 1 if the node belongs to the healthy brain tissue,
>
> and 0 if the node has a fixed boundary condition.
>
> The reason for including the spherical coordinates in the feature set was for wanting to include the magnitude of the applied force. Moreover, the idea of including the spherical coordinates of the force was inspired by the features introduced in the work by Tonutti et al., [2017] which has been cited in the paper. Lastly, the dimensions of the last linear layer is 63 by 3.
>
> **Possible Points for Improvement**
>
> We appreciate your input regarding possible areas for improvement. To enhance the understandability of the method section, we will change the notation of the weights in Equations 1 and 2. Moreover, aside from the evaluation metrics we are using to assess the performance of our approach following the previous works in the literature, we can also add the median and mode of the error values to the new revision of the paper.
>
> **Thank You**
>
> We hope we have addressed your questions and concerns and are happy to further discuss with you during the author-reviewer discussion period.
>
>
> **References**
>
> Michele Tonutti, Gauthier Gras, and Guang-Zhong Yang. A machine learning approach for real-time modelling of tissue deformation in image-guided neurosurgery. Artificial intelligence in medicine, 80:39–47, 2017.

---

### Official Review · Reviewer_BHqj · 2022-07-11

**Rating:** 5
**Confidence:** 4
**Soundness:** 2 fair
**Presentation:** 3 good
**Contribution:** 3 good

**Summary:**

The authors present a series of graph neural networks (GNNs) trained on finite element analysis (FEA) data to model deformations of brain tissue during surgery. The authors evaluate these networks on two separate datasets, and compare against baseline FEA results. The results show that the GNN framework is capable of milisecond inference on GPU hardware, whilst still retaining a clinically feasible level of accuracy.

**Questions:**

1. In section 5.1, you quote the time taken for the model to produce predictions on a GPU & CPU. Do you have the time taken to produce the  results using the original baseline data (i.e. the calculated FEA results)? Including this, as well as a quote of the speedup factor would be very useful as one of the main motivations of the model is the relative compute efficiency boost you would gain from this method.

2. You mention in section 5.4 that you lacked "accessibility to other studies’ data and/or implementation". Did you attempt to contact the authors to get access to the aforementioned datasets & implementations to validate your method against theirs?

3. You also mentioned in section 5.4 that in some cases you lacked "sufficient computational power to use their data/models". Could you elaborate on this please?

**Limitations:**

The authors appropriately address the limitations of the study, including the lack of validation datasets present within other studies, and the inherent limitation of using FEA as a baseline.

**Strengths And Weaknesses:**

**Strengths**

The paper is clearly written and concise. The work presents a novel approach to modelling FEA data using a GNN whilst clearly outlining why a GNN would be a suitable architecture for this problem. The introduction and related information clearly outlines the prior work and motivations relevant to this topic.

**Weaknesses**

The validation of the method is fairly limited, as is discussed in the limitations section 5.4. Without direct comparison to other methods data it is difficult to draw meaningful conclusions about the performance relative to similar methods.

In addition, the data used for validation is generated from finite element simulations, which is not as rigorous a source of comparison as compared to Lorente et al. (2017) which measure tissue deformations from ex-vivo data.

---

> ### Author Response · Authors · 2022-08-02
> **Response to Reviewer BHqj**
>
> We would like to thank you for your time, constructive criticism and positive comments on the clarity of our work and for acknowledging its suitability and effectiveness in approximating tissue deformation. We now address the questions and concerns you have raised in your review.
>
> **Original Baseline Data**
>
> Approximating tissue deformation caused by applying forces using the finite element method (FEM) on the CPU takes an average of 11.5099 ± 6.8004 seconds, with the minimum and maximum computation times of 5.4191 and 32.3798 seconds, respectively. On the other hand, PhysGNN takes an average time of 1.8294 ± 0.0322 on the CPU, as mentioned in the paper. Therefore, PhysGNN is capable of approximating the results 6.29 times faster. As recommended by you, we will add this information to the future revision of our paper to further emphasize the motivation of our research.
>
> **Comparison to Other Studies**
>
> As mentioned in our paper, one of the limitations of our study was having limited computational resources, which was a deciding factor in selecting the dataset for our work. Mesh data occupies significant memory when the mesh consists of many nodes or when a large number of FEM simulations are being used for training the model(s). We encountered difficulties when attempting to benchmark our models against different datasets because of the inability to reproduce the exact mesh structure in other studies for using different softwares for mesh generation and mesh optimization. We further faced challenges when wanting to implement their technique on our dataset either because of memory shortage or incompatibility of the nature of our dataset with theirs (e.g. different inputs or only computing deformation on a subset of data). Authors whose datasets were not publicly available and contained a large number of nodes—e.g. 300,000 nodes—were not contacted as we became aware of how large a mesh as well as how many finite element simulations we could use in our study with respect to our resources based on trial and error. Moreover, generating finite element simulations on very large meshes for at least 4000 times to create a rich dataset posed a different challenge for demanding a lot of computation power.
>
> We would like to note that while we were unable to reproduce the mesh data used in other studies for the reasons we discussed above, we followed the same evaluation methods employed in other works to draw meaningful conclusions.
>
> **Thank You**
>
> We hope we have addressed your questions and concerns and are happy to further discuss with you during the author-reviewer discussion period.

---

### Author Response · Authors · 2022-08-02
**General Response**

We would like to thank all reviewers for their time and constructive criticism. We are encouraged that all reviewers felt our paper was well-written and easy to understand. We are also delighted that they have found our contribution significant and suitable for approximating soft tissue deformation upon prescribed forces. Finally, we are especially thrilled that Reviewer PQLo has recognized and acknowledged the generalizability and applicability of our work to more than one area. We will answer the questions raised by the reviewers below, and we are happy to further discuss with them during the author-reviewer discussion period.

---

### Meta-Review · Area_Chair_hmo1 · 2022-09-10

**Recommendation:** Accept
**Confidence:** Certain

**Metareview:**

The paper proposes a method for modeling soft tissue deformations using a graph neural network trained to approximate the solution of the more classic finite element method. The proposed method is significantly faster than the classic method and with sufficient accuracy, enabling the prospect of using the method to predict tissue deformation in image-guided neurosurgery.

All reviewers appreciated the novelty and utility of the proposed method. The biggest weakness in the study had to do with the limited validation performed. This limitation was clearly addressed by the authors both in the limitations section and in their response to the reviewers. Despite the limitation, the reviewers unanimously recommend acceptance.

**Award:**

No

---

### Decision · Program_Chairs · 2022-09-14

Accept